# Designing a Cross-Cultural Bridging Intervention to Increase Under-Served Immigrant Parents’ Engagement in Evidence-Based Online Parenting Programs: A Co-Design Study with Indian-Origin Parents in Australia

**DOI:** 10.3390/children12091158

**Published:** 2025-08-30

**Authors:** Sunita Bayyavarapu Bapuji, Ling Wu, Joshua Seguin, Patrick Olivier, Marie Bee Hui Yap

**Affiliations:** 1School of Psychological Sciences, Monash University, Clayton 3800, Australia; sunita.bayyavarapu@monash.edu; 2Action Lab, Monash University, Clayton 3168, Australia

**Keywords:** co-design, culture, migrant families, online parenting program, youth mental health

## Abstract

**Highlights:**

**What are the main findings?**
Indian-origin parents in Australia face multiple, interrelated barriers to engaging with parenting programs.Six design principles were co-developed with parents to guide the development of a cross-cultural bridging intervention that connects with existing parenting programs.

**What is the implication of the main findings?**
Cross-cultural and experience-based co-design approaches can effectively address cultural and practical barriers that limit immigrant parents’ engagement with parenting programs.Embedding specific strategies in parenting, both program design and outreach, may improve access, uptake and sustain engagement among immigrant families.

**Abstract:**

**Background:** One in seven youth experiences a mental disorder, accounting for 13% of the global disease burden. The family environment is a modifiable factor for the prevention of mental disorders. While evidence-based online parenting programs exist, engagement by immigrant families, such as Indian-origin families in Australia, remains low. **Objective:** To explore perceived barriers of Indian-origin parents and co-create strategies to build cross-cultural bridging interventions to increase their engagement in parenting programs. **Method:** A qualitative co-design method was used, and participants were selected using a set of inclusion criteria through a criterion-based sampling approach. Eight videoconference workshops were conducted with 23 Indian-origin parents living across Australia, incorporating scenarios, roleplay, and vignettes. Data were analysed using Braun and Clarke’s inductive coding approach. **Results:** One central theme and six design principles were developed. The central theme, *low engagement with parenting programs*, encompassed five factors that contribute to low engagement: ‘parenting programs’ is not a concept in India; limited awareness of parenting programs available in Australia; lack of time to engage in parenting programs; misalignment between parenting program content and real-world parenting challenges; and an ‘I know how to parent’ mindset. The six design principles were: acknowledge culture shock and acculturation; use a collaborative approach; include content specific to immigrant parents and children; adopt cross-cultural perspectives; use short and interactive bilingual pedagogic tools; and use focused dissemination and marketing. **Conclusions:** This study’s findings formed the foundation for developing a cross-cultural bridging intervention to connect Indian-origin parents with existing online parenting programs.

## 1. Introduction

### 1.1. Youth Mental Health Problems

Globally, an estimated 13% of youth (i.e., children and adolescents) aged 10 to 19 years have diagnosable mental health problems (MHPs) within a 12-month period, such as depression and anxiety disorders [1]. MHPs are among the leading causes of illness and disability among youth, which may persist into adulthood if not diagnosed and treated early [2]. While there are several factors responsible for or related to the development of youth MHPs [3], the family environment is one of the key modifiable factors [4]. Specifically, various systematic reviews [5,6,7,8] have demonstrated that parental behaviours, such as less parental warmth and higher levels of inter-parental conflict, are associated with higher levels of youth MHPs. These findings underscore the importance of parenting programs in addressing youth MHPs. Moreover, burgeoning evidence suggests that modifications to parenting behaviours, with the help of parenting programs, could prevent or reduce youth MHPs [9].

### 1.2. Preventive Parenting Programs for Youth Mental Health

Researchers, practitioners, and organisations focusing on preventing or treating youth MHPs have developed several face-to-face and online parenting programs [10,11]. For example, *existing evidence-based online parenting programs* (hereafter referred to as parenting programs) such as Partners in Parenting (PiP) [12] and Parenting Resilient Kids (PaRK; now PiP-Kids) [13] have demonstrated efficacy in improving parenting factors associated with reduced risk for youth depression and anxiety, parental distress and parenting self-efficacy [12,13,14,15]. However, despite their accessibility, online parenting programs have seen low uptake among some persistently underserved subgroups, including immigrant-origin families [16]. This is particularly important given evidence that immigrant youth experience significant mental health challenges. For instance, a large population-based study in British Columbia examined diagnostic records of children and youth aged 0–19 years over a 20-year period, and found that second-generation youth—those born in the host country—had a higher prevalence of diagnosed mental disorders than first-generation youth, with reported prevalence rates of mental disorders at 5.94% of second-generation children aged 6–12 years compared to 4.30% among their first-generation peers [17]. In contrast, non-immigrant youth in the same age group had a mental disorder prevalence of 9.20% [17]. While these figures for second- and first-generation youth appear lower than the global 12-month prevalence estimates of 13% for youth mental health disorders [1], they reflect cumulative diagnostic prevalence based on administrative health records and may not capture the full extent of need. Gadermann and colleagues [17] concluded that their findings highlight the differences in the prevalence of diagnostic mental disorders among both generations of immigrant youth and non-immigrant youth and suggested that further investigation is required into how cultural differences and barriers to accessing available resources and services may be contributing to these differences. Along the same lines, the low uptake of parenting programs by culturally diverse populations suggests a significant gap between the availability of parenting programs and their relatability and acceptability to these communities.

## 2. Background

### 2.1. Context of Indian-Origin Population in Australia

The Indian-origin population, primarily voluntary economic migrants, is one of Australia’s 200-plus culturally and linguistically diverse (CALD) communities [18]. CALD communities include people born overseas or with at least one overseas-born parent who speaks a language other than English, which is Australia’s official language. As of June 2022, Indian-born people living in Australia were estimated at 753,520 (2.9% of Australia’s population) [19], excluding those Indian-origin youth under the age of 15 years born in Australia, commonly known as second-generation Indian Australians. Notably, Indian-origin immigrant families often experience unique acculturative stressors, including parent–child acculturative conflict, which exacerbate parenting challenges and increase the risk of youth MHPs [20,21,22]. However, there remains a dearth in our understanding of the underlying factors contributing to Indian-origin parents’ non-engagement in parenting programs. Involving these families in identifying factors and effective strategies to increase their participation in parenting programs is important for supporting the well-being and positive development of their youth and helping families thrive and contribute meaningfully to Australian society.

### 2.2. Rationale for a Cross-Cultural Bridging Parenting Intervention

There are several possible approaches to increase the engagement of immigrant-origin parents in parenting programs, including: (1) co-developing new parenting programs specifically for each cultural group [23]; (2) meticulously adapting parenting programs for each cultural group [24]; and (3) developing a bridging intervention that motivates parents to engage in existing parenting programs.

Co-developing an entirely new parenting program for every cultural group, while seemingly thoughtful, is not a feasible solution for theoretical and practical reasons. The literature indicates that many of the modifiable parental protective and risk factors associated with youth mental health in non-immigrant families are largely similar to those in immigrant families [25,26]. Parenting programs grounded in decades of peer-reviewed research already target many of these factors and have demonstrated robust evidence in improving parenting outcomes in general populations across many high-income English-speaking countries [9,10,27]. Therefore, it makes sound theoretical sense to find ways to build on such programs rather than creating new programs. Hence, instead of creating entirely new programs for immigrant families of diverse cultures, a more strategic approach may be to identify strategies to increase their engagement in available, well-established parenting programs.

However, since these programs are designed for the general population, they do not address some of the unique challenges immigrant families face. These challenges, such as acculturation struggles and navigating differing cultural expectations, are highly relevant to immigrant parenting experiences and are associated with immigrant youth mental health outcomes [25]. Hence, existing programs for the general population need to be supplemented to adequately address these additional factors that are pertinent for immigrant families.

Yet, meticulously adapting parenting programs to make them more culturally suitable to each immigrant group may not be the optimal solution. Cultural adaptations typically focus on surface-level modifications, such as language translation or incorporating culturally relevant examples, while overlooking deeper issues such as the intersecting effects of social, cultural and historical factors on parenting behaviours that are critical for immigrant families’ mental health and well-being [28]. Moreover, adapting programs for every cultural group is costly and time-intensive [29] and given the diversity of immigrant populations in high-income countries, it is unlikely that such an approach could equitably meet the needs of all groups who make high-income countries their home.

In contrast, we propose a cross-cultural bridging intervention approach, which may offer a more effective and sustainable alternative. A bridging intervention can engage immigrant parents by focusing on and addressing their immediate concerns, such as acculturation-related parenting struggles [30,31]. In addressing their most salient needs, the intervention can establish credibility with immigrant parents and build on that by then raising their awareness of the relevance of broader parenting strategies already embedded in evidence-based programs. As such, a bridging approach can provide targeted support to help immigrant parents connect with and see the relevance of universal parenting programs. Such an approach can not only enhance accessibility but may also promote equity by providing a viable, scalable solution for increasing the engagement of diverse groups of immigrant parents in parenting programs [28,32].

### 2.3. Theoretical and Methodological Frameworks

This study draws on three key frameworks to develop a bridging intervention for Indian Australian immigrant parents: the Double Diamond (DD) framework, the Experience-Based Co-Design (EBCD) method, and the Capability, Opportunity, and Motivation-Behaviour (COM-B) model. Each framework plays a distinct role in guiding the research design, participant engagement and interpretation of findings. The DD framework provides the overall structure, EBCD ensures cross-cultural relevance, and the COM-B model strengthens the design and development of the intervention and its potential for real-world impact. These are briefly discussed below.

#### 2.3.1. Double Diamond Framework: Structuring the Research Process

The DD framework [33,34] provides a structured approach to systematically explore problems and develop solutions through a four-stage process: the research phase (Discover), the synthesis phase (Define), the ideation phase (Develop), and the implementation phase (Deliver). A diverse range of prior research has used this framework to develop a telehealth peer support program for caregivers of individuals with dementia [35]. DD is particularly well-suited to culturally complex design problems, as it enables an iterative engagement with diverse stakeholders, such as Indian-origin families, allowing for an in-depth exploration of parental factors and cultural practices, synthesis of insights into culturally grounded problem definitions, and the co-creation of socially responsive and contextually appropriate solutions. As this study requires a systematic approach to understand the limited engagement of Indian-origin parents in parenting programs, and to develop a bridging intervention to empower, support and increase their engagement in parenting programs in Australia, the DD framework was deemed to be an optimal option due to its capacity to guide culturally responsive and inclusive intervention design.

#### 2.3.2. Experience-Based Co-Design: Centering Lived Experiences

The EBCD method originated from participatory action research and ensures that solutions are developed collaboratively with end-users, considering their specific context and needs, which increases the likelihood of their acceptance and use [36]. A recent systematic review indicates that EBCD is associated with greater intervention engagement [37]. EBCD also facilitates the incorporation of cultural values, norms, and parenting practices into the design process by centring the voices of culturally diverse participants in both the discovery and co-design phases. This makes it particularly valuable for designing cross-cultural interventions where cultural sensitivity and contextual relevance are essential for engagement and effectiveness. As such, EBCD seems to be a promising method to partner with Indian-origin parents in the design of a cross-cultural bridging intervention to increase their engagement in parenting programs that are relevant and accessible in Australia. Thus, we used the EBCD method within the DD framework to define the problem and ideate strategies by fostering trust and co-ownership among Indian-origin parents.

#### 2.3.3. COM-B Model: Understanding and Addressing Barriers

While DD provides the overall framework and EBCD a method for co-developing lived-experience centred solutions, the COM-B model [38] offers a behaviour change lens to mitigate the barriers experienced by Indian-origin parents and identify strategies to engage them in parenting programs. The COM-B model posits that behaviour is influenced by three core elements: capability (knowledge and skills), opportunity (social and physical environments), and motivation (reflective and automatic processes). The COM-B model has been successfully applied in various contexts, such as to improve hearing aid usage in adults [39], and its use is now beginning to emerge in parenting research [40]. Therefore, this approach to behaviour change is promising in helping to identify crucial factors that inhibit or facilitate engagement and inform the design of the bridging intervention, which will align with the cultural and contextual realities of Indian families in Australia.

### 2.4. Current Study Objectives

Following the DD framework, this study aimed to collaborate with Indian-origin parents, utilise EBCD to gain a deeper understanding of their knowledge of parenting programs, identify barriers to their engagement in such programs and lay the foundations for designing bridging interventions to motivate Indian immigrant families to engage in various parenting programs that cater to parents of children across different age groups. Additionally, it sought to apply the COM-B model to discuss the design principles for developing a bridging intervention.

## 3. Methods

### 3.1. Ethical Considerations

All procedures involving human participants in this study were conducted in accordance with the ethical standards of the Monash University Human Research Ethics Committee (MUHREC) and the 1964 Helsinki Declaration and its later amendments. Ethical approval was obtained from the Monash University Human Research Ethics Committee (MUHREC 31860) on 20 April 2022.

### 3.2. Guiding Frameworks and Models

Using the DD framework (Figure 1), in the *Discover* phase, we conducted a systematic review of the literature on modifiable immigrant parenting factors associated with youth mental health [26] and a qualitative study to understand Indian parenting practices related to youth mental health in Australia [30]. In the initial *Define* phase, findings from these two studies were synthesised to gain insights and assess the needs of immigrant-origin families. To complete the *Define* phase, the current study followed the EBCD method to work with Indian-origin immigrants to define the problem of limited engagement in parenting programs and identify principles to guide the design of the bridging intervention (*Develop* phase). The design principles will be discussed using the COM-B theoretical framework to describe how the bridging intervention can be designed to guide and bridge immigrant families to engage in parenting programs (*Delivery* phase). Note that this study focused on identifying barriers and intervention design principles to support engagement; the development and delivery of the actual bridging intervention lie beyond its scope.

### 3.3. Participant Sample, Recruitment and Demographics

The participant sample for this study included Indian-origin adults living in Australia, aged 18 years or more, currently parenting children aged 10 to 18 years and having the ability to engage in conversation to some extent in English and any of the Indian languages. First, following the qualitative study recruitment strategy, the first author posted the study information flyer on numerous Facebook groups that cater to Indian-origin people living in specific cities (e.g., Melbourne, Sydney, Brisbane and Perth), and states in Australia (e.g., Victoria, New South Wales, Queensland and Western Australia). Care was taken to promote the study to those speaking various Indian languages or following specific religions. Second, parents who participated in the preceding qualitative study that the authors conducted [30] and expressed interest in future research were also invited to participate in this study.

Interested parents contacted the first author, who emailed them the Participant Information Sheet and Consent Form. Parents who provided consent were invited to scheduled workshops via Zoom and were given gift vouchers of $40 per hour as a token of gratitude for sharing their experiences and insights in workshops.

The workshop participants were from regional and metropolitan areas across four states of Australia. They originated from ten different states and territories of India. Participants spoke English and at least one Indian language. All were living with their spouses and children. Table 1 presents other characteristics of the workshop attendees.

### 3.4. Positionality

The positionality of the research team played an important role in shaping the design, implementation, and interpretation of this study. As co-creators of the study’s meaning, the positionality of researchers is crucial. All the authors have immigrant origins and live in Australia. The first author is a female PhD candidate who conducted the workshops, including participant recruitment, data collection and analysis. She is a mother herself and reared her child as an immigrant. Her identity as an Indian-origin parent of a young adult was explicitly shared with participants to establish rapport, build trust, and create a culturally safe environment that encouraged open and active engagement. She has a background in nursing, undertook training in qualitative research, and has been working as a researcher in the health system for the last 15 years. These personal and professional experiences enabled her to relate closely to participants’ stories, while also requiring continuous reflexivity to minimise assumptions during analysis.

Three of the four co-authors are academics with expertise in qualitative research and co-design. The last author is a parent and has vast expertise in parenting research. However, except for the first author, the other co-authors are not of Indian background. This difference in cultural background was acknowledged during data interpretation and addressed through regular reflexive discussions to check for potential bias and to ensure that interpretations remained grounded in the participants’ perspectives. None of the authors had any pre-existing relationships with the participants.

### 3.5. EBCD Workshops and Data Collection

This study utilised the EBCD method to lay the foundations for designing a bridging intervention to motivate Indian-origin families to engage in parenting programs.

The first steps in developing the workshop guide and format were (i) synthesising the systematic review and qualitative study findings and (ii) reviewing the literature and theoretical frameworks on EBCD, as well as intervention development and adaptation.

Workshops with Indian-origin parents were conducted between June 2023 and December 2023. The first author facilitated eight workshops, consisting of three sets, which were attended by two to six participants and lasted 90 to 120 min each. One participant attended all three sets of workshops, three attended two sets of workshops, and 19 attended one of the sets of workshops. To accommodate participants from across different time zones within Australia and to provide flexibility, the workshops were conducted online via Zoom.

The Appendix A file outlines the steps followed to conduct all workshops. The first set of workshops explored participants’ knowledge of parenting programs and barriers to engagement. The second set of workshops identified and discussed strategies for inclusion in the bridging intervention. The third set of workshops sense-checked the findings with participants and prioritised strategies. Finally, the research team conducted a design ideation process to develop design principles for the bridging intervention.

### 3.6. Data Analysis

The first author audio-recorded all workshops and transcribed them using original Zoom transcriptions and made corrections. Further, she took observation notes after each workshop. Subsequently, data from transcriptions were uploaded to NVivo 14 software for analysis. The author’s notes and sticky notes from the online whiteboard were also collated.

The collected workshop data were analysed using Braun and Clark’s [41] six-phase thematic analysis, following an inductive coding approach to name, organise, and interpret the content. First, all transcripts and other notes were read in detail for familiarisation and codes were generated to capture the engagement barriers and strategies to engage in interventions. Second, codes were examined to identify the theme and factors for low engagement and subsequently searched for themes to develop design principles. These were then reviewed, defined, and named, and the analysis was finalised. To ensure credibility, the first author held meetings with co-authors throughout the data collection and analysis process to discuss the developed themes and to draft design principles. Disagreements were resolved through discussion and consultation with the last author, who is a subject matter expert. Member checking was carried out in the form of sense-making workshops when the data from the first six workshops were analysed.

## 4. Results

The workshops’ data led to the development of one central theme with five interrelated factors (Figure 2) and six design principles (Figure 3). All quotes are labelled with a unique identifier to maintain participants’ anonymity. The identifiers include the source (W# = workshop number) and the participant type (F = father; M = mother).

### 4.1. Central Theme: Low Engagement with Parenting Programs

As presented in Figure 2, the central theme, ‘low engagement with parenting programs’, encompasses five underlying factors that represent key barriers that explain the low engagement of Indian-origin parents in parenting programs.

#### 4.1.1. Factor 1: Parenting Program Was Not a Concept in India

Some parents reported that they never encountered terms such as “parenting program” during their time in India. One father mentioned, “*Parenting program isn’t a concept that’s accepted or widely used in India, at least when I left the country*” (W1F4). This suggests a cultural gap in the understanding and acceptance of structured parenting support systems. Additionally, most mothers indicated that parenting practices were passed on intergenerationally, and thus, their parenting practices closely mirrored those of their own upbringing, relying on the approaches of their grandparents, parents, aunts, and uncles. One mother said, *“We are more comfortable raising our kids the way we were raised”* (W1M3). This reliance on traditional practices underscores the limited recognition of parenting programs within Indian cultural contexts.

#### 4.1.2. Factor 2: Limited Awareness of Parenting Programs Available in Australia

Most parents were unaware of the availability of parenting programs. As one mother mentioned, “*I didn’t come across any Indian parents so far, who was talking about this online parenting program to me”* (W1M1). Another mother mentioned that she attended a session offered at their children’s school, as a teacher mentioned X program [name of the program used in role-play]. Pointing to another parent in the workshop, another mother said: “*he knows about the program… since he is in a bigger city. It might be more very widespread to that population, and I am from a smaller city”* (W2M1). Parents’ awareness of parenting programs is contingent upon the availability of these programs within their children’s schools or local communities. This suggest that parents tend to become aware of parenting programs through interactions in their community, including educators, school representatives, or other Indian parents who highlight and discuss the existence and benefits of these programs.

#### 4.1.3. Factor 3: Lack of Time to Engage in Parenting Programs

Parents who were aware of parenting programs mentioned that, as immigrants, they often struggle to find the time to participate in such initiatives. Most parents reported that their primary focus was to secure the basic necessities for their families. As articulated by one father, the priority was to “*put a roof over your family’s head and to ensure they can put the bread on the table”* (W2F1). The absence of domestic help in their new environment, which was commonly available in India to manage household chores, also compounded this challenge. Fathers indicated that they must balance work commitments, including the demands of a typical 9-to-5 job or shift work, and support their spouse with household responsibilities. As a result, parents mentioned that they often have limited time for activities like reading parenting materials or participating in structured programs. After fulfilling work and family obligations, they preferred to unwind with leisure activities instead of dedicating time to learning about parenting.

#### 4.1.4. Factor 4: Misalignment Between Parenting Program Content and Real-World Parenting Challenges

Parents who engaged with the content of parenting programs reported concerns about the applicability of these programs to their experiences as immigrants. Some noted that the advice seemed tailored for “*Australian good kids, idealistic kids”* (W3F2) and often fell short when addressing real-life parenting challenges. One mother described, “*I looked at online [resources]. I did try all of that stuff, but it didn’t. It did not help me or her [daughter]*” (W1M2), referring to her daughter’s difficulties making friends at the new school.

Another father criticised the advice provided, describing it as “idealistic” and impractical for their situation; he expressed frustration with guidance that suggested simply listening to a crying child, stating, “*What does it even mean? So, I think it’s not [helpful]. It was very idealistic, fanciful reading*” (W3F1). These statements reflect parents’ difficulties in perceiving the value and relevance of research-based advice with the realities of their busy work–life schedules, which often require quick, practical and context-relevant solutions.

#### 4.1.5. Factor 5: “I Know How to Parent” Mindset

Some parents expressed that they knew how to parent their children effectively in a new country. Parenting was often viewed as an inherited set of practices passed down through generations, and there was a sense of confidence in continuing with familiar approaches. One mother stated, “*I personally don’t look for a parenting program to guide in the ways I am parenting my kids*” (W1M4).

Among those who were aware of parenting programs, some parents noted that cost was a barrier. Interestingly, they reported that programs were too expensive to engage in, but at the same time, programs are not valued when they are offered for free. As one father said, “*there’s another mindset issue that we need to tackle within. And that is something that’s for free… people don’t value it. Right? So, it’s only when you pay for something [that people value a program]*” (W2F1). This suggests a culturally shaped perception that the value of a service is tied to its cost, where free programs may be perceived as having lower quality or credibility. This belief further complicates their engagement with parenting resources, even when those resources are made available for free.

### 4.2. Design Principles

To address the barriers and challenges captured under the central theme, workshop participants collaboratively brainstormed to help develop design principles (i.e., suggestions and recommendations; Fu et al.) [42] to engage Indian immigrant-origin parents in parenting programs through the bridging intervention. The six resulting design principles were: (1) acknowledge culture shock and acculturation; (2) use a collaborative approach; (3) include content specific to immigrant parents and children; (4) adopt cross-cultural perspectives; (5) use short and interactive bilingual pedagogic tools; and (6) use focused dissemination and marketing (Figure 3).

#### 4.2.1. Acknowledge Culture Shock and Acculturation

All parents emphasised the profound culture shock they and their children experienced after moving to Australia and the challenges of navigating two distinct cultures. One parent said, “*When a family makes a decision to come into a foreign country…the children get stuck in between the Indian culture and the new culture”* (W4M3)*,* stressing how children often feel *“sandwiched*” between the differing cultural expectations. Parents in this study perceived that these cultural differences contributed to their children gravitating more toward their peers and the new environment. As the same parent mentioned, “*They get jelled into that more*” (W4M3), making it difficult for parents to emphasise the equal importance of Indian culture.

Parents also expressed discomfort and concern about their children adopting the behaviours they see in children in Australia, which reflect different cultural values and boundaries from their own. One mother described how her 6-year-old daughter’s interactions with boys, such as hugging and kissing, are seen as normal in Australian schools but were culturally shocking for her. She said, “*This is something… a culture shock for me. I really don’t want my girl to indulge in that level of activity… it’s pretty normal for them [non-immigrant], but it’s really different for me*” (W5M2).

Similarly, parents had to navigate conflicting cultural norms of parenting (e.g., paying their children for doing household chores vs. treating such chores as a reflection of the children being an integral part of the family), and learning and helping their children follow social norms (e.g., what is considered an appropriate birthday present for their child’s classmate) in the new environment. As one parent mentioned, “ *With the Aussie population, if you do these dishes, then I’ll pay you a dollar, and if you do that, we’ll pay you a dollar. It never happens in my house. Doing dishes that’s chore and that is expected of my son. We don’t pay*” (W1M3). Another parent noted how this difference became a source of conflict when their son expected to be paid like their friends, prompting difficult conversations about family values. These experiences indicate how cross-cultural differences can create parenting challenges, especially when children’s expectations are shaped by their interactions with their peers.

These reflections underscore that immigrant-origin families require support to deal with culture shock and navigate the acculturation process—balancing their home country’s values with the cultural norms of their new environment. One parent mentioned that this process is “*very, very time-consuming*” and requires significant “*contribution from the parents, from the grandparents*” (W5F1) to help children appreciate both cultures equally. Additionally, parents suggested, “*Maybe this program can be blended in a way that people can more easily talk about it [culture shock]*” (W4M2), and because it is an “*issue that every migrant faces [and] goes through it [acculturation process]*” (W4M1), although they may be unaware of the terminology.

#### 4.2.2. Use a Collaborative Approach

Parents valued the co-design approach, where intervention planning was undertaken collaboratively with them, rather than just asking them to participate in a pre-designed program. They appreciated being involved in shaping the intervention and saw this as crucial for ensuring relevance and improving engagement among Indian-immigrant families. One parent commented, *“I’m glad that you are working on it [intervention] with us. It will really help people [other immigrant parents]”* (W8F2). Another parent shared, *“Most of the topics [topics outlined for bridging intervention] are good and definitely would help a large [immigrant] audience for parenting, and I think this is required”* (W7F2). These reflections suggest that parents endorse the inclusion of parents early in the development process, believing that the resulting intervention would be more likely to be seen and felt as appropriate and meaningful to the target population.

In addition, some parents expressed a desire for the planned intervention to take a proactive approach to help immigrant parents build confidence and parenting skills before challenges arise. One parent also suggested prompting intervention participants to reflect on their own parenting goals through guided questions, to support early engagement: *“What do you think would be a good approach to bring up your kids?… and then you identify [in the intervention] what they are and then show them challenges and attract them [to engage in the intervention]”* (W8F1).

Together, these perspectives point to a clear preference for collaborative and empowering interventions that engage immigrant parents early, build on their strengths, and increase their confidence in managing parenting in a new cultural environment.

#### 4.2.3. Include Content Specific to Immigrant Parents and Children

Tailoring content to the specific experiences of immigrant parents ensures that the advice is practical and relatable. This approach can help bridge the gap between theoretical concepts and culturally contextualised real-world applications. Parents mentioned the importance of providing content that addresses their children’s unique needs in adjusting to school and social settings in a new culture, which can minimise feelings of loneliness. One parent indicated that the intervention could help with understanding that “*This is important for your kids to be mingling around with the Australian population and to know their culture*” (W1M1). They also highlighted the need for resources on the prevention of bullying, transitioning between two cultures (e.g., language and food-related boundaries) and building support for acculturation.

Additionally, parents expressed concern about helping their children understand the risks associated with certain behaviours, with one parent pointing out that “*kissing, sex, alcohol, and drugs seem to be common topics, and there are boy gangs*” (W3F2). This concern of parents emphasises the importance of equipping parents with the tools to have these conversations with their children effectively. Furthermore, one parent shared concerns about diverse family structures and not being equipped to help their children understand, “*My kid comes home and asks me, ‘Why are there two mums picking up that kid and two dads picking up this kid?*” (W3M2). This highlights the need for content that supports parents in understanding and guiding their children through discussions about family diversity and acceptance.

Parents indicated that parent-specific content is essential regarding culturally rooted tendencies toward over-protectiveness that can arise from a desire to safeguard children in an unfamiliar cultural environment. *“It’s such a common problem with every parent because the kids want to go out and eat, and you are conflicted in your mind, and you really don’t know what to do about it. Like you want to be liberal, and at the same time want to hold on to your roots [not eat meat]”* (W6M2). They emphasised the need for the intervention to help parents explain cultural differences and build resilience in their children while fostering a deeper understanding of their children’s emotions. Parents acknowledged the importance of granting autonomy to their children, acknowledging that this can be challenging in immigrant families where traditional values often prioritise parental control and oversight. They expressed the need for guidance in such situations and finding a balance between academic focus and encouraging leisure and sports activities, which may differ from cultural expectations of educational achievement.

Additionally, parents recognised the necessity of avoiding fear-based parenting strategies, which are sometimes perpetuated in conservative cultural communities. One parent mentioned, “*Parents need to empower their kids, not scare them*” (W4M4). By addressing these culturally embedded practices, the intervention can better equip parents to adapt their parenting approaches in a way that respects both their heritage and the new cultural context in which they live.

Furthermore, parents expressed the need to understand and explain sexual orientation to their children and the value of proactively mingling with parents from different cultural backgrounds to learn their cultural nuances. By addressing both child-specific and parent-specific content, a bridging intervention can foster a comprehensive understanding that empowers immigrant families to navigate the complexities of raising children in a new cultural context.

#### 4.2.4. Adopt a Cross-Cultural Perspective

Parents acknowledged the need for the bridging intervention to incorporate cross-cultural perspectives to better support immigrant families like theirs. They shared that recognising the unique experiences and challenges they face would make the programs more practical and relatable. One parent stated, “*It is important to combine elements of both cultures*” (W2M2), highlighting the significance of integrating their heritage with their new environment. They also expressed the value of learning from others, with one parent sharing, “*I learned from an Australian friend to reinforce routines and give less guidance in daily tasks that my son has to do*” (W2M1). Parents suggested that the intervention should be tailored to reflect their realities and focus on strategies that blend both cultural norms. Additionally, they highlighted the need for content that encourages cross-cultural interactions. One parent suggested resources that explain “*how the program can help them mingle with the Australian population*” (W5M2), indicating a desire for guidance on how immigrants can better relate to and connect with locals.

#### 4.2.5. Use Short and Interactive Bilingual Pedagogic Tools

Parents who knew about the online parenting program stated, “*It was mostly text… I’m not sure if there was any video or audio to support those [parenting] concepts*” (W2F1), indicating a preference for visual and interactive materials. Parents emphasised the importance of incorporating short and interactive bilingual tools into the bridging intervention, such as videos and reels they see on social media. This is because educational resources in their own language with English subtitles or vice versa would help them feel more comfortable engaging. They also indicated that these formats could better attract Indian-origin parents and cater to their busy schedules.

Parents also mentioned pictures and animated resources such as role plays, stories and series. As one parent suggested, “*Some videos like Bluey [TV show]*” (W3M1) would be beneficial. Some other parents added the value of “*animated role play*” (W3F3) and “*animated stories…very powerful*” (W1M2). Further, they expressed a desire for content that reflects their experiences, with one parent noting, “*If there is like a short series kind of stuff, like where it’s actually focusing on Indian parents… an animation character who, like a kid who is of Indian ethnic, and then they go through the same process*.” (W1M4). In contrast, some parents indicated a preference for videos featuring real-life people of Indian origin sharing their lived experiences of challenges and how they adapted. Overall, the feedback from parents underscores the necessity of developing content that is informative, engaging, and accessible for parents navigating the complexities of raising children in a cross-cultural environment.

#### 4.2.6. Use Focused Dissemination and Marketing

Parents identified three key aspects of focused dissemination and marketing to increase engagement with existing parenting programs and the bridging intervention: how it is messaged, where it is promoted, and who should be specifically targeted.

First, parents emphasised the importance of framing the intervention as preventive and for all parents—not only for those whose children are already facing problems. This approach helps to reduce stigma and encourages proactive participation. One parent explained, “*It needs to be promoted in a way that we don’t wait for kids to have anxiety or depression. So, you need to work through the program beforehand*” (W3F3).

Second, many parents suggested that dissemination efforts should take place across settings frequented by immigrant families, including schools, kindergartens, general practitioner clinics, local councils, Centrelink, libraries, and community and religious centres. As one parent suggested, “*Marketing of that video has to be done by somebody from the kindergarten or the school, or maybe the temples where most of the Indian population goes to, or the community centres where they go to are important*” (W1M1). Another added, “*I think if one person in the community knows [about this intervention] and they can use word of mouth*” (W2M2). A few parents also recommended immigration offices as a point of initial program dissemination for newly arrived families. Practical tools such as factsheets and fridge magnets were proposed as everyday visual cues. In addition, they suggested utilising mass media channels such as TV, radio, and parent forums as effective means of outreach.

Third, some parents stressed the importance of reaching fathers, in particular recognising them as a group less likely to engage without targeted efforts. They suggested that employers could play an active role in encouraging this group of caregivers’ participation in the bridging intervention.

Taken together, these insights suggest that a multi-level, cross-culturally attuned marketing strategy is needed to effectively raise awareness of both existing programs and the bridging intervention among immigrant-origin parents.

## 5. Discussion

To the best of our knowledge, this is the first study to develop one central theme—*low engagement with parenting programs* encompassing five barriers to engaging Indian-origin parents in parenting programs—(1)‘parenting programs’ is not a concept in India; (2) limited awareness of parenting programs available in Australia; (3) lack of time to engage in parenting programs; (4) misalignment between parenting program content and real-world parenting challenges; and (5) an ‘I know how to parent’ mindset.

Findings also delineated six design principles—(1) acknowledge culture shock and acculturation; (2) use a collaborative approach; (3) include content specific to immigrant parents and children; (4) adopt cross-cultural perspectives; (5) use short and interactive bilingual pedagogic tools; and (6) use focused dissemination and marketing to address these engagement barriers.

While the results are presented thematically to reflect the voices of Indian-immigrant parent workshop participants more naturally, the COM-B model offers a valuable framework for interpreting the key findings and shaping the application of the resulting design principles. Accordingly, this discussion uses the COM-B model to contextualise how the proposed cross-cultural bridging intervention may enhance engagement among immigrant-origin parents by addressing capability, opportunity, and motivation. Although the discussion focuses on design principles, this focus is grounded in the earlier findings about low engagement, which served as the foundation for identifying barriers and enablers. These insights were then used to develop practical, context-specific design principles to increase engagement. This interpretation is supported by prior research and informs key implications for research, practice and policy.

### 5.1. Psychological Capability

The principle of ‘acknowledge culture shock and acculturation’ directly addresses the psychological capability aspect of the COM-B model by revealing an important psychological barrier immigrant-origin parents face in adapting to Australian cultural norms. This principle highlights the unique need of immigrant-origin parents for a cross-cultural bridging intervention that addresses culture shock and acculturation challenges, which are not covered in existing parenting programs. Parents’ accounts of feeling “*shocked*” and “*overwhelmed*” by Australian norms reflect the psychological demands of balancing their culture with new social-cultural expectations. Research shows that such acculturation challenges can strain family well-being, as differing rates of cultural adaptation between parents and children may lead to conflicts [43]. These differences in adaptation rates can exacerbate parents’ feelings of shock and overwhelm, likely reducing their sense of self-efficacy about their parenting approaches [44].

Therefore, as suggested by parents, incorporating lived experiences and structured discussions on acculturation within the bridging intervention could help normalise these challenges and provide parents with practical strategies to build psychological resilience. This, in turn, can support their own well-being and their children’s mental health. These insights align with Berry’s model of acculturation [45], which emphasises that successful adaptation involves both maintaining key aspects of one’s culture while engaging with the host culture. Integrating this perspective into the intervention design could help parents make sense of their own experiences and develop the awareness and skills needed to navigate culture shock and acculturation challenges—and to support their children in doing the same. By incorporating cross-culturally relevant examples and addressing common acculturation challenges, the cross-cultural bridging intervention can strengthen families’ adaptation and confidence while also offering a familiar and supportive pathway into existing parenting programs, such as PiP and PiP-Kids [12,13], which immigrant parents may otherwise find unfamiliar or difficult to engage with, despite their availability.

### 5.2. Physical and Social Opportunities

The principle of ‘short and interactive bilingual pedagogic tools’ directly addresses the physical and social opportunities component of the COM-B model by highlighting how physical and social opportunities together could play a vital role in facilitating immigrant parents’ participation in a bridging intervention. Parents consistently emphasised that short, interactive, bilingual, and visually engaging resources, such as videos and animations featuring cross-culturally relevant content, could make the intervention more relatable, especially when they reflect both Indian and Australian parenting norms. These forms of content not only offer physical accessibility by catering to different literacy levels and learning preferences but also foster social familiarity by embedding everyday parenting experiences within culturally recognisable narratives. For example, storytelling to depict cross-cultural parenting challenges and family dynamics may allow parents to see aspects of their own lives reflected in the material, creating a sense of cultural resonance and emotional connection. These strategies can address varying literacy levels, promote inclusivity, and accommodate parents’ preferred learning modes [46,47,48].

Importantly, parents wanted access to materials in both English and their native languages, increasing accessibility and comfort in learning [49,50]. This linguistic and cross-cultural relevance can make the bridging intervention feel more familiar and supportive, fostering a sense of belonging while acknowledging their bicultural parenting realities. Tailored interventions that incorporate cross-culturally meaningful perspectives are more likely to be perceived as credible and personally relevant, potentially leading to greater engagement with recommended strategies [51,52].

By drawing from evidence-based parenting programs such as PiP and PiP Kids [12,13], and embedding familiar and cross-culturally resonant elements, the bridging intervention could function as a practical and meaningful ‘taster’. In doing so, it may help immigrant parents see the applicability of these universal programs to their own parenting challenges and feel more confident exploring them further. Rather than replacing these programs, the bridging intervention can enhance their reach and impact by increasing the perceived relevance of the programs to immigrant parents.

In addition to content, the principle of ‘use focused dissemination and marketing’ directly addresses the physical and social opportunity components of the COM-B model by indicating that promotional methods should be embedded in parents’ existing physical and social environments. Meaning, parents recommended promoting the bridging intervention through familiar and trusted channels such as schools, places of worship, community hubs, and social media platforms like Facebook and WhatsApp. These are natural settings where they already seek parenting advice. Research supports this approach, as community-based dissemination increases awareness, trust, and uptake of parenting interventions, particularly among underserved populations [53,54,55]. Framing the bridging intervention as a brief, practical, and cross-culturally responsive resource that addresses immediate parenting concerns in a new country could further lower the barriers to participation.

### 5.3. Reflective Motivation

The principle of ‘use a collaborative approach’ directly addresses the reflective motivation component of the COM-B model by suggesting that a collaborative bridging intervention approach has the potential to encourage immigrant parents to critically consider their parenting practices in relation to a new cultural context. This reflective process can, in turn, promote greater openness to adapting their behaviours and building new parenting skills. Parents’ suggestions underscore an ardent desire to be actively involved in shaping the intervention, reflecting their motivation to learn and understand. By engaging parents as co-creators, the intervention promotes a sense of agency and ownership, positioning them as empowered contributors to their children’s development. Research supports the effectiveness of empowerment-focused parenting programs in improving engagement and long-term outcomes, such as change in parenting self-efficacy and positive parenting [56,57,58].

Involving immigrant parents early in the intervention planning process, such as through co-design and input on cross-culturally relevant content, could help establish credibility and increase the acceptability of the resulting bridging intervention. When paired with practical and interactive tools, this collaborative approach could also promote a sense of ownership of the bridging intervention among immigrant families. Importantly, co-designing content that reflects shared cultural values and everyday parenting experiences can elicit real-life examples that resonate with parents, prompting them to reflect on their own parenting journeys. This reflective process, rooted in familiarity, can deepen emotional engagement and enhance motivation to participate. This, in turn, may motivate parents to engage more actively, share their experiences, and raise awareness of the intervention within their networks. Ultimately, such proactive and inclusive strategies may help immigrant parents feel more confident and better prepared to navigate the challenges of bicultural parenting in their host country, such as Australia.

### 5.4. Psychological Capability and Reflective Motivation

The principle of ‘include content specific to immigrant parents and children’ addresses both psychological capability and reflective motivation components of the COM-B model by emphasising the importance of the bridging intervention to be more specific about addressing unique challenges faced by immigrant families, such as bullying and discrimination. This contextualised problem-solving approach could bridge the gap between theoretical concepts and real-world applications, making the content more practical and relatable. By equipping parents with targeted tools and cross-culturally relevant strategies, the intervention can enhance their psychological capability, particularly their knowledge, confidence, and readiness to support their children’s integration into the local culture. These findings align with previous research suggesting that immigrant parents benefit from specific, skills-based resources that not only help them navigate the cultural context but also build confidence and motivation to support their children’s development in a new country [55,59].

In addition, parents expressed concern about helping their children manage risks associated with alcohol, drugs, and unsafe sexual practices and emphasised the need for the bridging intervention to provide guidance on navigating these sensitive conversations with their children. This aligns with studies by Pantin and associates [60] and Lee and associates [61], which highlight that immigrant parents require specific tools to prevent unsafe behaviours in their children.

Furthermore, parents highlighted a common theme in immigrant parenting literature: the challenge of balancing traditional protective practices with fostering independence [62,63,64]. Therefore, the bridging intervention should include guidance on supporting their children’s autonomy in ways that resonate with their cultural values. For instance, parents could be encouraged to engage in discussions with their children about selecting their extracurricular activities or choosing subjects for higher education. Such guidance can enhance parents’ reflective motivation, encouraging them to engage more actively in the bridging intervention. This cross-cultural bridging intervention would briefly address these topics and then direct parents to the existing parenting programs for more detailed strategies to address these challenges, as well as raise their awareness that these programs also provide strategies on other topics of interest and relevance, such as helping their children deal with bullying and adopting healthy habits and behaviours.

### 5.5. Strengths and Limitations

This research study provides a unique contribution to the current immigrant and parenting literature by including Indian-origin parents in the research process, a population that is fast growing in high-income countries globally but is notably underrepresented in parenting research, particularly in Australia. A key strength of this study is the use of the EBCD method within the DD framework with Indian-origin parents, and most of them, without hesitation, shared their unawareness of available online parenting programs. Additionally, all voiced their concerns about barriers to engaging in such programs, even if they are freely available and provided several strategies to improve Indian-origin parents’ participation.

This EBCD study was conducted to help fill the gap in the literature on Indians in Australia. The COM-B model was also used to interpret and discuss design principles for a cross-cultural bridging intervention. However, the limitations of the study should be taken into consideration. First, all Indian immigrant parent participants were recruited using Facebook groups. Thus, the findings may suffer from self-selection bias. While this remains a limitation, Facebook is widely used among Indian immigrant communities in Australia and was considered a practical and feasible method for reaching a diverse and engaged sample of parents in this formative, exploratory study.

Second, all parents were part of intact families, tertiary educated, employed, financially stable, lived in regional and metropolitan areas and had English proficiency. While these participants’ characteristics represent the majority of Indian migrants in Australia [65], the self-selection nature of this study may have unintentionally excluded Indian parents who are less engaged, less focused on parenting, or more affected by significant disadvantages, such as non-internet users, those with lower English literacy levels, unemployed, going through family-related challenges and socially isolated. Additionally, this study did not capture the perspectives of first- and second-generation Indian-origin young adults who are currently parenting or are prospective parents. Future research could adopt a more inclusive sampling strategy to ensure the representation of such groups. Additionally, the EBCD method may have favoured those familiar with Zoom videoconferencing and group discussions and those with outgoing personalities. As a result, all these factors may limit the generalizability of the findings, especially concerning preferences for technology-based and interactive interventions.

Finally, the Indian-origin population in Australia is highly diverse, encompassing multiple languages, religions and cultural backgrounds. Even though potential participants were approached through various languages and religious Facebook groups, several languages and religious and cultural groups were underrepresented. The findings may overrepresent the experiences of Indian-origin parents with greater socioeconomic privilege, which limits the breadth of perspectives captured. While this study laid the foundation to address common barriers to engagement in parenting programs, future research is needed to explore the distinct experiences of sub-groups within this population, to ensure that parenting interventions adequately cater to their needs.

### 5.6. Implications for Research, Practice and Policy

Although EBCD methodology in research with ethnic and minority communities is gaining recognition [66,67], this study is among the first to engage Indian-immigrant parents in the pre-development process of a behaviour change intervention, a bridging intervention from an acculturative perspective. Importantly, a recent systematic review indicates that EBCD is associated with greater intervention engagement [37].

One implication of this study is that the EBCD approach is valuable for engaging immigrant families in the co-design of interventions that are more likely to resonate with their specific needs and preferences. However, a notable limitation, as previously mentioned, is that this study’s participants were limited to parents who migrated from India and had children who accompanied them from India (commonly referred to as first-generation or naturalised) or were born in Australia (commonly referred to as second-generation). Previous research emphasised the significance of including the voices of both first and second-generation young adults [30], particularly since some are currently parenting or are prospective parents. Therefore, an important recommendation for future research is to involve this demographic in co-designing interventions and evaluation efforts actively.

Research shows that parental involvement in home, educational, and social settings plays a crucial role in addressing acculturative challenges, strengthening intergenerational relationships, and supporting both effective parenting and youth mental health outcomes [68,69]. Recommend engaging two to three-generation immigrant-origin families can therefore ensure that bridging interventions are not only cross-culturally relevant but also responsive to evolving generational dynamics within immigrant families. The EBCD approach offers a promising way to facilitate this engagement in future research, fostering collaboration across generations and cultural contexts. Therefore, it is recommended to use EBCD approaches in future research in this area.

Beyond Indian-origin families, the approach and findings from this study have broader implications in Australia. The barriers and facilitators identified through this process provide valuable insights for parenting and mental health services, policymakers, and system designers aiming to tailor support for newly arrived and established immigrant families. Co-developing bridging interventions with diverse immigrant groups can help ensure these initiatives reflect shared lived experiences, improve accessibility and trust, and ultimately increase engagement in existing parenting programs. Thus, it is recommended that collaborative partnerships with these communities be undertaken in future research to support parenting and youth mental health across diverse cultural settings.

## 6. Conclusions

This study provides valuable insights into Indian-origin parents’ perceptions of evidence-based online parenting programs and the specific barriers that limit their engagement in these programs. It also co-developed design principles for a cross-cultural bridging intervention that is intended to increase Indian-origin parents’ engagement in existing online evidence-based parenting programs. The findings offer a practical foundation for integrating the developed design principles, such as adopting cross-cultural perspectives and the use of bilingual pedagogic tools into interventions that Indian-immigrant parents can relate to and for delivering these interventions to reach Indian families across Australia.

The unique contribution of this study is applying an experience-based co-design method with an underrepresented Indian-origin community in Australia to inform the development of a cross-cultural bridging intervention. These findings have important implications for designing future interventions, particularly for researchers planning to develop cross-cultural online bridging interventions to support immigrant families and connect them with existing evidence-based parenting programs. Findings are also relevant to mental health service providers and organisations that assist immigrant and refugee families in settling into new countries. Ultimately, implementing such interventions has the potential to improve immigrant parents’ engagement with parenting programs, strengthen their parenting confidence and enhance mental health support systems for immigrant populations in Australia and beyond.

## Figures and Tables

**Figure 1 children-12-01158-f001:**
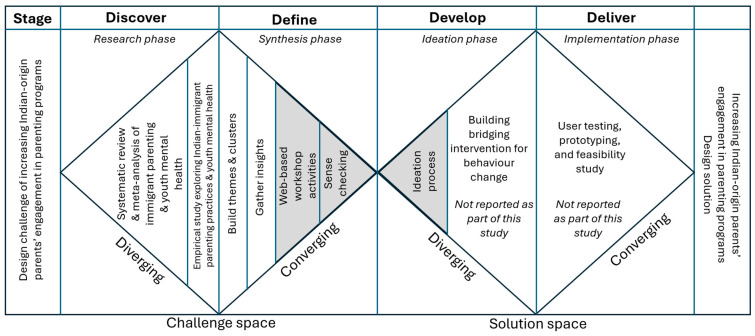
The Double Diamond framework adapted to develop a bridging intervention. Note: The shaded areas pertain to the current study.

**Figure 2 children-12-01158-f002:**
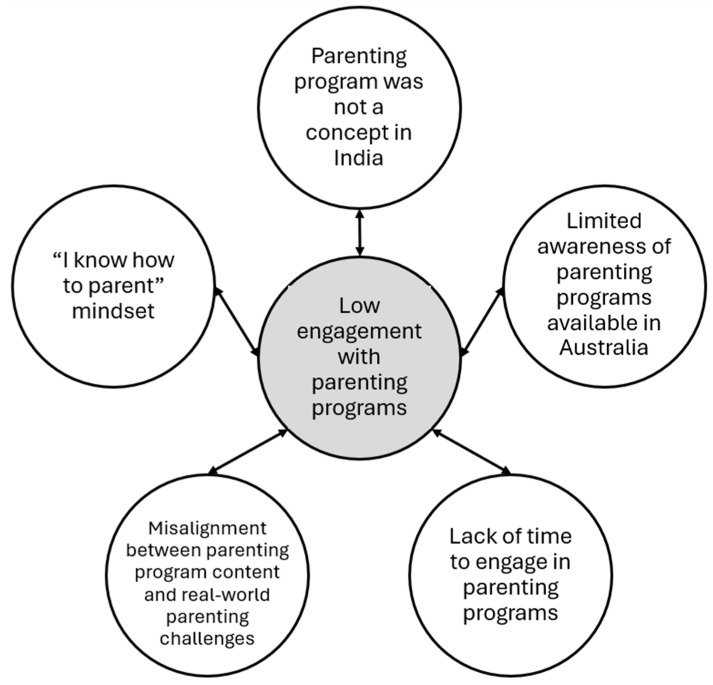
Thematic map showing the central theme of ‘Low engagement with parenting programs’ and five underlying factors.

**Figure 3 children-12-01158-f003:**
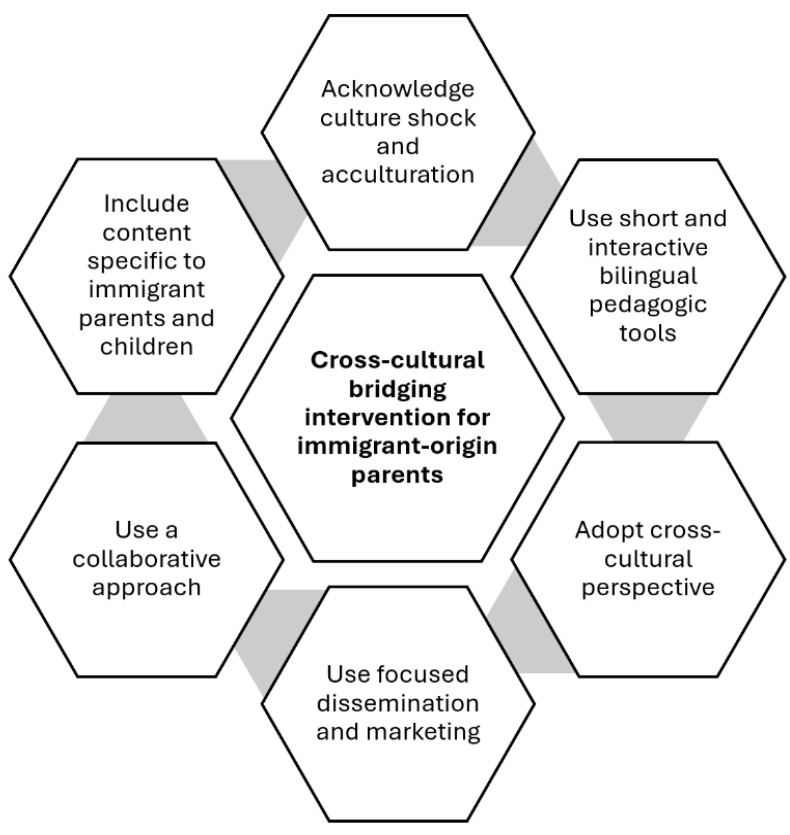
Design principles for a cross-cultural bridging intervention for immigrant-origin parents.

**Table 1 children-12-01158-t001:** Characteristics of the Indian-origin parents who participated in the workshops.

Characteristic	Description	Count	%
Indian-Origin Parents (*n* = 23)			
Parent type	Mother	13	57
	Father	10	43
Tenure in Australia	5–10 years	11	48
	11–20 years	10	43
	20+ years	2	9
Whether lived in other countries prior to migrating to Australia	Lived in other countries besides India	6	26
	Not lived in any other country besides India	17	74
Highest education qualification	Graduate	12	52
	Postgraduate	11	48
Current employment status	Employed	23	100

## Data Availability

The data that support the findings of this qualitative co-design study cannot be made available due to ethical considerations and the need to maintain participant confidentiality. Although identifying information has been removed, the data cannot be shared to ensure that participants’ identities remain protected.

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
