# Peer review of "Designing a Cross-Cultural Bridging Intervention to Increase Under-Served Immigrant Parents’ Engagement in Evidence-Based Online Parenting Programs: A Co-Design Study with Indian-Origin Parents in Australia"

_children, 2025, doi:10.3390/children12091158_

Round 1
Reviewer 1 Report
Comments and Suggestions for Authors
Thank you for the opportunity to review your work.
- Abstract
Your abstract should also indicate the research approach used, e.g qualitative approach. You should also indicate how the participants were selected and what sampling methods you used. Other than that, your abstract is well-written and captures all the important elements of the manuscript.
- Introduction
Your manuscript seems to be missing the introduction section. May I kindly suggest that you do an introduction for your manuscript? Alternatively, consider changing your background to the introduction section. The background discussion can start on 1.3, where you discussed the Context of the Indian-Origin Population in Australia. `Should you go ahead with this suggestion, then you can delete the subheadings 1.1 and 1.2 and then readjust your numbering accordingly.
- Background
It is good that this section presents information in a manner that demonstrates relevancy and appropriateness to the topic. This section also indicates your solid understanding of the relevant issues. Furthermore, well done on clearly articulating the theoretical frameworks followed. Kindly consider the above comment regarding moving some of the information in this section to the introduction section.
- Methodology
Your methodology is well written and comprehensive. However, more clarity could have been provided on the population used for the study, you alluded to specific suburbs, cities, and states in Australia without clearly identifying on naming them. Is there any reason for this? The study area also appears to be too broad. Which specific states in Australia did your study focus on, and how were these geographic areas identified? Your manuscript appears to be silent on the sampling methods used; what were the inclusion and exclusion criteria? More information could have been provided on how the data analysis was done. You should also consider discussing the ethical implications of the study. Other than these highlighted issues, your methodology is solid, well done.
- Results
Your results section is well-written and supported with quotes from participants. It is clear that the collected information was relevant to the objectives of the study.
- Discussion
Your discussion section is well written and linked to the findings of the study. It is good that you also supported your points, views, and statements with relevant literature sources. The influence of your theoretical frameworks in drafting this section is also evident. Well done.
- Conclusion
Your conclusion is adequate and sound. Consider moving 4.6 “Implications for research, practice and policy” from the discussion section to the conclusion section. Recommendations could be further illuminated.
- Overall Recommendation
- Accept after Minor Revisions: The paper can, in principle, be accepted after revision based on the reviewer’s comments.
Reviewer 2 Report
Comments and Suggestions for Authors
Thank you for the opportunity to review the manuscript. The manuscript addresses an important issue and has many merits. However, I have some concerns and comments that should be addressed to improve the manuscript.
1) This study centers on the Indian-origin population in Australia—a focus that is both strategic and necessary. However, it is essential to recognize the considerable diversity within this group. India, as a subcontinent, encompasses a wide range of languages, religions, and cultural traditions. As such, the term “Indian-origin” may overlook the distinct experiences of individuals.
2) The introduction is quite dense and extended, which could hinder clarity and accessibility for readers at the outset.
3)The study combines exploratory goals—such as understanding parents’ knowledge and identifying barriers to engagement—with the more complex task of designing an intervention, which can be demanding within a single study. Additionally, targeting parents of children across a wide age range, from early childhood to adolescence, may introduce considerable variability, as parenting needs differ significantly by developmental stage.
4) Although the co-design approach offers valuable in-depth insights, the sample size of 23 parents is relatively limited. Additionally, all participants were highly educated (holding graduate or postgraduate degrees) and employed at the time of the study. As such, they may not reflect the broader Indian-origin population in Australia, which also includes individuals with lower educational attainment, those who are unemployed, or recent immigrants. Consequently, the findings may disproportionately represent the perspectives of a more socioeconomically privileged subgroup within the community.
5) The recruitment strategy, though innovative, depended largely on individuals already active on social media or those with prior research participation. This may have resulted in a sample biased toward people who are more engaged, connected, or receptive to research compared to the broader population. Additionally, the workshops involved only a small number of participants.
6) The data analysis section is somewhat concise. While the authors mention using Braun and Clarke’s inductive coding approach and holding meetings with co-authors to enhance credibility, providing additional detail would strengthen the rigor of the study. For instance, clarifying specific steps taken to ensure trustworthiness, such as whether member checking with participants was conducted, and explaining how disagreements among co-authors were resolved during theme development would offer greater transparency and robustness to the analysis process.
7) The discussion uses the COM-B model to interpret the findings, which is a strength. However, the connection between a specific design principle and a COM-B component could be even more explicit in some places. For instance, while it is implied, the text could more clearly state, "The principle of 'short and interactive bilingual pedagogic tools' directly addresses the physical and social opportunity component of the COM-B model by..." This would further strengthen the theoretical grounding of the discussion.
8) The "Conclusion" section is rather brief and reads more like a summary of the abstract than a compelling closing statement. It reiterates the study’s purpose and overall relevance without emphasizing the key takeaways. A more impactful conclusion would succinctly highlight the most important finding and their unique contribution, followed by a strong, forward-thinking statement on how bridging interventions could transform mental health support for immigrant communities.
Reviewer 3 Report
Comments and Suggestions for Authors
Dear authors,
Title – clearly describes the content of the article.
Abstract – in the Methods section, briefly state only the number of participants and their demographic characteristics.
Keywords – please arrange them in alphabetical order.
Introduction – the introduction thoroughly describes the issue and introduces the reader to the problem; however, at times it seems overly detailed, causing the focus to be lost. Therefore, I suggest condensing subsections 1.3, 1.4, and 1.5, focusing only on the essential facts. For example, in subsection 1.5, instead of 2–3 examples, one illustrative example per model would be sufficient.
I also believe the sentence “By designing a cross-cultural bridging intervention, this study seeks to maximise the relatability, acceptability, and impact of parenting programs to support parents of children of all ages, from early childhood to adolescence, who are growing up in Australia.” is redundant, as this has already been elaborated earlier.
Methods – Table 1 should be condensed and provide only a brief description of participants’ demographic characteristics. If age is not categorized in the results, consider presenting some measure of central tendency (mean, median).
It would be good to shorten this section to the essential points. I suggest reducing the part describing the workshops, as there is no need to explain each one individually. Also, the description of Zoom’s functions is overly technical; it would suffice to say simply “Zoom with technical tools.”
Similarly, detailed descriptions of the design are not necessary here; if based on previous studies, please refer to them.
In short, describe the sample, the procedure for conducting workshops briefly, and the data analysis; the rest, if too detailed, as mentioned also for the introduction, can be overwhelming for the reader.
Consider whether Figures 2, 3, 4, and 5 need to be included in the article if they are referenced in the Methods section.
Results – The results are well described and clearly presented, providing a comprehensive understanding of the study’s findings.
Discussion – In the section concerning recommendations for further research, it would be beneficial to clearly specify these recommendations.
References – The references are appropriate, relevant, and current, supporting the study well.
Round 2
Reviewer 2 Report
Comments and Suggestions for Authors
Thank you for your revision. The revised version addressed most of my comments and concerns.
Reviewer 3 Report
Comments and Suggestions for Authors
Dear Authors,
Thank you for your detailed clarifications and the valuable revisions you have made. I have no further comments or concerns regarding your article. I believe it is suitable for publication.
Best regards